# In Vitro Evaluation of the Safety and Efficacy of *Cibisatamab* Using Adult Stem Cell-Derived Organoids and Colorectal Cancer Spheroids

**DOI:** 10.3390/cancers17020291

**Published:** 2025-01-17

**Authors:** Victor Anstett, Elisa Heinzelmann, Francesco Piraino, Aline Roch, Antonius Chrisnandy, Maxim Norkin, Virginie Garnier, Krisztian Homicsko, Sylke Hoehnel-Ka, Nathalie Brandenberg

**Affiliations:** 1Doppl SA, 1015 Lausanne, Switzerland; victor.anstett@doppl.ch (V.A.); elisa.heinzelmann@doppl.ch (E.H.); francesco.piraino@doppl.ch (F.P.); aline.roch@doppl.ch (A.R.);; 2Department of Oncology, CHUV, 1066 Lausanne, Switzerland; maxim.norkin@chuv.ch (M.N.); krisztian.homicsko@chuv.ch (K.H.)

**Keywords:** T cell bispecific (TCB) antibodies, adult stem cell-derived organoids, co-culture, preclinical safety and potency assessment

## Abstract

This study introduces an advanced 3D assay using colorectal cancer spheroids and adult stem cell-derived, healthy human organoids to evaluate the efficacy and safety of *Cibisatamab*, a bispecific antibody targeting carcinoembryonic antigens (CEAs) on cancer cells and CD3 on T cells. Our approach integrates live imaging and cytotoxicity analysis with standardized workflows, providing precise and reproducible insights into therapeutic effects. The findings reveal that *Cibisatamab* effectively targets high-CEA-expressing cancer cells and is dose-dependent on target, off-tumor binding and killing on non-cancerous cells. By capturing complex tumor-immune interactions and variations in therapeutic responses, this model offers a valuable tool for advancing immunotherapy research and predicting potential safety liabilities in clinical trials.

## 1. Introduction

Immunotherapies have emerged as powerful approaches in cancer treatment, aiming to harness and direct the body’s own immune cells to identify and attack malignant cells. These therapies can be broadly categorized into two classes: active and passive.

Active immunotherapies stimulate the body’s own immune system to enhance antitumor responses, either by counteracting immunosuppressive mechanisms, as seen with immune checkpoint inhibitors (ICIs), or by amplifying immune activity through immunostimulatory pathways, as seen with T cell bispecific (TCB) antibodies or bispecific T cell engagers (BiTEs) [1,2,3]. In contrast, passive immunotherapies directly deliver immune components, such as chimeric antigen receptor T cell (CAR-T) therapy and tumor-infiltrating lymphocyte (TIL) therapy. These approaches do not rely on the patient’s immune system to initiate the response but instead deliver agents that target and attack cancer cells directly [4].

Among active immunotherapies, ICIs have gained importance by blocking key immunosuppressive molecules, such as programmed cell death 1 (PD-1) or its ligand PD-L1, enabling cytotoxic T cells to attach tumor cells more effectively [5]. The approval of *lpilimumab* by the Food and Drug Administration (FDA) in 2011 marked a significant milestone in ICI-based therapies [6]. *lpilimumab* targets cytotoxic T-lymphocyte-associated protein 4 (CTLA-4) to potentiate the T cell-mediated antitumor immune response. Despite its success in extending survival in metastatic melanoma patients, ICIs like *lpilimumab* are associated with immune-related adverse events (irAEs) and limited efficacy in tumors with insufficient tumor-infiltrating lymphocytes (TILs) [7,8].

Passive immunotherapies, such as CAR-T cell therapy, have also demonstrated remarkable potential. CAR-T cell therapy involves genetically engineered patient T cells to express chimeric antigen receptors (CARs) that recognize tumor-associated antigens (TAAs). This modification enables T cells to selectively eradicate cancer cells [9]. While this approach is effective, the process of isolating, modifying, and expanding CAR-T cells is complex, costly, and time-consuming.

An emerging alternative within immunotherapies is the use of multi-specific immune engagers and TCB antibodies. Multi-specific immune engagers can target multiple TAAs simultaneously [10,11], while TCB antibodies typically bind a single TAA on the tumor cell while, in parallel, engaging the CD3 epsilon chain of the T cell receptor (TCR) complex, bridging T cells to cancer cells and triggering a potent immune response [3].

The first TCB, *Blinatumomab*, was approved by the FDA for the treatment of hematological cancers that originate from B cells [12]. Since then, nine TCBs have been approved, with seven for hematological cancers and two for solid tumors [13]. *Blinatumomab*, a CD19 BiTE, belongs to the class of Fc-free TCBs. Even though this class of TCBs shows a high killing potency, there is a disadvantage of its clinical application due to its short half-life and the need for continuous infusion. In contrast, newer IgG-based TCBs exhibit longer half-lives, improving clinical practicality [14]. The carcinoembryonic antigen (CEA) TCB is a member of the IgG-based bispecific antibody class and targets the CEA glycoprotein (also known as CEACAM5 or CD66e), which is anchored in the cell surface through glycosylphosphatidylinositol (GPI) [15]. CEA is overexpressed in various solid tumors, including colorectal, pancreatic, gastric, and non-small-cell lung cancers. In normal tissues, CEA expression is low and restricted to the apical surface of glandular epithelia in the gastrointestinal tract, limiting the accessibility to therapeutic antibodies [16,17]. However, even low expression levels of the target antigen in healthy tissues can lead to on-target, off-tumor toxicity [18].

Assessing the safety and efficacy of CEA TCBs in preclinical models poses unique challenges. Conventional animal models are unsuitable, as the anti-CEA arm of the antibody lacks cross-reactivity with species such as cynomolgus monkeys, and there are no CEA orthologs in rodents or other species for preclinical safety evaluations [19].

To address these limitations, advanced in vitro models, such as three-dimensional (3D) cell cultures and spheroids, offer a promising alternative. Spheroids, derived from cancer cell lines, replicate some aspects of tumor architecture and cell–cell interactions better than two-dimensional (2D) cultures, though they lack cellular diversity and complex tissue architecture [20]. In the last two decades, organoids have emerged as powerful in vitro models. Derived from patient samples, organoids can retain the genetic and phenotypic features of both healthy and tumor tissues, providing a more physiologically accurate model for drug testing [21,22]. Importantly, organoids enable the evaluation of on-target, off-tumor toxicity within the same donor-derived normal and cancer tissues. The use of 3D spheroids and organoids, thus, enhances the predictive power of preclinical studies, helping to bridge the gap between conventional 2D in vitro experiments and clinical applications, while reducing reliance on animal models that may lack relevant cross-species targets.

In this study, we present the 3D redirect lysis (3D-RDL) assay, a novel approach for the preclinical assessment of immune cell engagers, such as CEA TCBs. This assay evaluates the potency of CEA TCBs using 3D cancer spheroids and healthy organoids co-cultured with human peripheral blood mononuclear cells (hPBMCs) from healthy human blood samples. By capturing the complexity of tumor–immune interactions, the 3D-RDL assay overcomes the limitations of traditional 2D in vitro systems and provides critical insights into both efficacy and safety. Specifically, we assessed the CEA TCB *Cibisatamab*, a 2:1 bispecific antibody targeting CEA on tumor cells and CD3ε on T cells, using cancer spheroids and healthy organoids [23]. This approach enables a more precise characterization of the therapeutic window and dose determination for such antibodies, which is essential for minimizing on-target, off-tumor toxicity.

## 2. Materials and Methods

### 2.1. Healthy Human Peripheral Blood Mononuclear Cells (hPBMCs) Isolation and Culture

hPBMCs were isolated from Buffy Coats (Don du sang—Epalinges, Switzerland) using Ficoll gradients and SepMate isolation tubes (85450, STEMCELL Technologies, Vancouver, BC, Canada). hPBMCs were cultured in complete RPMI (cRPMI) composed of RPMI + 10% Fetal Bovin Serum (FBS) + MEM Non-Essential Amino Acids (1X) + 1% Pen Strep + HEPES (1X) + Sodium pyruvate (1X).

### 2.2. Human Primary Organoids and Cell Lines

Tissue samples and annotated data were obtained and experimental procedures were performed within the framework of the non-profit foundation Human Tissue and Cell Research (HTCR) including the informed patient’s consent [24]. The generation and storage of organoids from these biopsies were approved by the Cantonal Ethics Committee (CER-VD AO_2021-00076) and conducted by Doppl SA in Lausanne, Switzerland. Biopsies were collected from a total of 7 patients (age 61 ± 16, sex 71% females). Biopsies were collected from normal adjacent tissue (NAT) upon surgical tumor resection of patients suffering from malignancies of colorectal, ileal, or pancreatic origin. Human intestinal crypts were isolated from rectum, colon (transversal, ascending and appendix), and small intestine (jejunum and duodenum) biopsies following previously established protocols [25]. The stem cells present in the crypts allowed the establishment of rectal, colon, and small intestinal organoids. For organoid cultures, we used a revised cell culture growth medium inspired from the group of Sato, containing advanced DMEM/F12 supplemented with GlutaMAX™ (35050-038, Thermo Fisher Scientific, Waltham, MA, USA), HEPES (15630-056, Thermo Fisher Scientific), penicillin/streptomycin (15140-122, Thermo Fisher Scientific), and B-27 Supplement without Vitamin A (12587-001, Thermo Fisher Scientific) [26]. The organoids were maintained and expanded as follows: cultured in a humidified atmosphere of 5% CO_2_ at 37 °C, the full medium was changed every 2 days, and the cells were passaged every 3–4 days.

MKN-45 (ACC 409, DSMZ) and DLD-1 (ACC 278, DSMZ) were cultured in the medium recommended by the supplier in a humidified atmosphere of 5% CO_2_ at 37 °C.

### 2.3. Organoid Characterization

The rectum organoid line and small intestinal (Small Intestine_2) organoid line used in this study were closer characterized through Immunofluorescent staining (Appendix A) and bulk RNA sequencing (Appendix A). 

Immunofluorescent staining: Prior to Immunofluorescent staining, organoids were collected, washed once with PBS, and fixed with 1 mL of 4% paraformaldehyde (PFA) for 30 min at RT. After three 10 min PBS washes, the fixed organoids were then stored at 4 °C for any immunofluorescence analysis. The respective organoids were blocked with blocking solution, containing 10% donkey serum and 0.2% Triton in PBS, for 3 h at RT and incubated with primary antibodies (MUC2: anti MUCIN2, 555926 BD Bioscience, San Jose, CA, USA; FABP1: anti-FABP1, AF1565 R&D Systems, Minneapolis, MN, USA; Ki67: ant-KI67, MA5-14520 Thermo Fisher Scientific; LYZ: anti-lysozyme, GTX72913 GeneTex, Irvine, CA, USA), diluted 1:200 in blocking solution, at 4 °C, overnight. The following day, the organoids were washed three times with PBS and subsequently incubated with secondary antibodies (donkey anti-mouse Alexafluor 568, A10037 Thermo Fisher Scientific; donkey anti-goat Alexafluor 488, A-11055 Thermo Fisher Scientific; donkey anti-rabbit Alexafluor 647, A31573 Thermo Fisher Scientific; and DAPI, 62248 Thermo Fisher Scientific) diluted 1:500 in blocking solution. Stained organoids were then washed three times with PBS and mounted on a glass slide with Fluoromount-G (Thermo Fisher Scientific). The fluorescence images of whole organoids were taken on an inverted confocal microscope (Olympus FV 4000) with a 20×/0.80 objective (2048 × 2048 pixels, 5.9535/µm). Image processing was performed using ImageJ (version 1.54K).

Bulk RNA sequencing: For bulk RNA sequencing, three biological replicates of the small intestinal organoid line and two biological replicates of the rectum organoid line were harvested, and the RNA was isolated. RNA was sent to a third-party contractor (Alithea Genomics, SA, Épalinges, Switzerland) for bulk RNA barcoding and sequencing (BRB-seq) and library preparation [27]. Raw reads were then demultiplexed and aligned to the human genome. The data were analyzed in R (4.4.1) with edgeR methods [28]. In brief, the lowly expressed genes were filtered out, and the counts were normalized and shown as log2 count per million (CPM). Data visualization was performed both in R and GraphPad Prism (version 10).

### 2.4. Allogeneic Co-Culture of hPBMCs

In the context of allogeneic co-culture experiments, target cells (MKN-45, DLD-1, small intestinal and rectal organoids) were incubated with varying amounts of hPBMCs at effector-to-target (E:T) ratios of 10:1, 5:1, or 2.5:1 for 72 h in the absence of treatment. During these preliminary experiments, one out of four healthy hPBMC donors exhibited mild spontaneous killing of target cells without treatment. To ensure the validity of the results, this donor was excluded from subsequent studies. These initial tests were conducted across all spheroid and organoid lines to establish baseline compatibility before assessing the killing potency mediated by treatment.

To validate the specificity of the readouts for evaluating target cell killing, hPBMCs (at the highest E:T ratio) and target cells were separately treated overnight with 0.8% Triton X-100. ATP content and lactate dehydrogenase (LDH) release were quantified to confirm the accuracy of the assay metrics.

### 2.5. Three-Dimensional Redirected Lysis (3D-RDL) Assay on Three-Dimensional Grown Target Cells

Organoids grown in solidified extracellular matrix were collected, enzymatically dissociated into single cell suspension, filtered, counted, and seeded in 96-well plates (Gri3D^®^96 Wellplate, 31 microwells per array, SunBioscience, Lausanne, Switzerland). Cells lines were collected from the flasks and processed the same way as organoids. Then, 2 days after target cell seeding, hPBMCs were seeded into cell culture flasks at 1 × 10^6^ cells per mL in cRPMI medium for overnight resting in an incubator at 5% CO_2_, 37 °C. At day 3, the hPBMCs were collected from the flasks, washed with FBS, counted, resuspended in target cell media, and added to the already-formed organoids/spheroids. *Cibisatamab* was added for 72 h to the wells at 6 different doses ranging from 0.001 to 100 µM (1:10 dilution factor). Each condition was tested in duplicates on 3 different hPBMC donors.

### 2.6. In Vitro Assessment of CEA-Mediated Killing Through Multiplexed Readouts

After 72 h, the supernatant of the 3D-RDL assay was homogenized and transferred to another 96-well flat bottom plate for LDH quantification, and the remaining volume was frozen at −20 °C for later cytokine analysis. LDH quantification was performed following the manufacturer guidelines (G1780, Promega, Madison, WI, USA). For live/dead quantification, cells were stained with 1 µM Calcein-AM and 2 µM Ethidium homodimer-1 (LIVE/DEAD Viability/Cytotoxicity Kit, L3224, Thermo Fisher, Waltham, MA, USA) for 1 h at 37 °C and imaged on an IN Cell Analyzer (GE Healthcare, Waukesha, WI, USA). The fluorescence signal was acquired over 20 Z-stacks (10 µm), deconvoluted, and analyzed in maximum projection. Through Doppl’s fully automated image analysis pipeline, organoids were automatically detected and segmented and the median fluorescence intensity and organoid/spheroid size were measured. Post-imaging, the supernatant was removed for ATP quantification following the manufacturer protocol (G9681, Promega). Absorbance and luminescence were measured on the multimode plate reader Spark (Tecan, Männedorf, Switzerland). For all the read-outs, lysis buffer (Promega) was used a positive control for total cell lysis.

### 2.7. T Cell Activation Analysis and Relative Target Expression by Spectral Flow Cytometry

After 72 h, hPBMCs were retrieved from the 3D-RDL assay, filtered, and transferred to a V-bottom 96-well plate and resuspended in cold PBS. Cells were stained in PBS with live/dead staining dye Zombie-NIR (423106, Biolegend, San Diego, CA, USA) and FcR blocking solution (130-059-90, Miltenyi, Bergisch Gladbach, Germany) for 20 min at 4 °C. Cells were washed in FACS Buffer (PBS + 2 mM EDTA + 2% FBS) and resuspended in FACS Buffer and extra-cellular staining cocktail, CD69-APC (310910, Biolegend), CD4-BV421 (300532, Biolegend), CD8-SparkUV 387 (100798, Biolegend), and CD25-PE (385608, Biolegend), for 30 min at 4 °C. Cells were washed twice and resuspended in FACS Buffer for spectral flow cytometry analysis.

To evaluate the relative target expression on single cell dissociated organoids and cell lines (DLD-1 and MKN-45), 1 × 10^5^ cells were seeded and stained as previously described with CD66d/e-AF647 (392806, Biolegend). All the samples were acquired using the EPFL Flow Cytometry platform on a 5-Laser Cytek-Aurora instrument, and FACS data were analyzed using FlowJo Software (version 10).

### 2.8. Soluble Granzyme B, TNF-α and IFN-γ Quantification and Analysis

Granzyme B, TNF-α and IFN-γ released in the supernatant were quantified using the LEGENDplex Multi-Analyte Flow Assay kit (BioLegend) according to the manufacturer’s instructions. Human CD8/NK Panel Detection Abs V02 was used to measure Granzyme B, TNF-α, and IFN-γ. Samples were acquired using a LSRFortessa (Becton, Dickinson, BD, Franklin Lakes, NJ, USA) flow-cytometer, and FCS files were analyzed with Biolegend’s LEGENDplex data analysis software (https://legendplex.qognit.com, assessed on 16 October 2024). Standard curve was generated from 8 standard doses with duplicates of each standard dose (Appendix A) When cytokine release was below the LLOD or above, the ULOD the value of the sample was set at LLOD or ULOD, respectively.

### 2.9. CEA Expression Levels

Data were extracted from the Protein Atlas website and displayed using the Prism 10 software. The collected data showed the normalized expression (nTPM) values from two different RNA-seq datasets: HPA and GTEx. They also showed an estimate of the highest protein expression score. Bulk RNA-seq data, generated at Doppl SA on different healthy organoids lines, were displayed as normalized log2 counts per minutes (Log2 CPM). Following enzymatic digestion described above, the single-cell suspensions of healthy rectal and intestinal organoids were FACS-stained and relative protein CEA expression levels were displayed by Prism 10. DLD-1 and MKN-45 cells were stained in parallel and acquired at the same time as the dissociated organoid lines.

### 2.10. Specific Lysis

The percentage of specific lysis was calculated by using the following equation:$$\%\;of\;specific\;lysis\left(ATP\right)=\frac{X-\max killing}{target\;cells\;only-\max killing}\times 100\%$$ (*X* = sample value; target cells only = target cells without treatment and effector cells; maximum killing = target cells treated overnight with 0.8% Triton-X).

### 2.11. Statistical Analysis

Statistical analyses and graph generation were performed using GraphPad Prism software (GraphPad Software, San Diego, CA, USA). Cytotoxicity upon *Cibisatamab* treatment was assessed by analyzing the effects of increasing doses on MKN-45, DLD-1, and small intestinal and rectal 3D cultures. An ordinary one-way ANOVA followed by Dunnett’s multiple-comparisons test was used to compare the mean of each condition to that of the reference condition (Target only). For the analysis of Granzyme B levels released in the supernatant following *Cibisatamab* treatment, the same statistical approach was applied, comparing the mean of each group to that of the reference group (100 nM on MKN-45).

Additionally, a two-way ANOVA with multiple-comparisons test was performed to compare the treated MKN-45 cell line/rectum organoids (reference column: 10 µM) for each column to the mean of the DLD-1 cell line/small intestinal organoids (Appendix A).

Only statistically significant differences are shown in the graphs, with significance levels denoted as follows: * *p* < 0.05, ** *p* < 0.01, *** *p* < 0.001, and **** *p* < 0.0001.

## 3. Results

### 3.1. Target Validation

First, we validated the expression of the target cell-surface glycoprotein CEA in healthy human tissue in two conventional adenocarcinoma cell lines (MKN-45 and DLD-1), as well as in patient-derived organoids from different organs (colon, rectum and small intestine) of the gastrointestinal (GI) tract. Protein expression data from the Human Protein Atlas indicated that CEA expression was highest in the rectum and colon, with lower levels observed in the small intestine (Figure 1B–D) [29]. Similarly, bulk RNA sequencing analysis of seven healthy GI organoid lines from Doppl SA’s biobank showed a comparable pattern, with the highest CEA expression detected in rectal and colon organoids compared to small intestinal organoids (Figure 1C).

We selected one rectal organoid line (Rectum) and one small intestinal organoid line (Small Intestine_2) for detailed characterization within this study (Appendix A). The rectal and small intestinal organoid lines expressed markers for key intestinal cell types, including Goblet cells (MUC2+), enterocytes (FABP1+), and Paneth cells (LYZ+), confirming their differentiation status and suitability as models for healthy rectal and intestinal tissue (Appendix A). Additionally, bulk RNA sequencing analysis revealed the activation of distinct canonical pathways and hallmarks characteristic of small intestine and rectum tissues, further supporting the tissue-specific identity of the organoid lines used in this study (Appendix A).

Next, we directly compared and analyzed relative CEA protein expression levels between the organoid lines and the adenocarcinoma cell lines MKN-45 and DLD-1 using flow cytometry (Figure 1D). The adenocarcinoma MKN-45 cell line exhibited significant overexpression of CEA, consistent with published results characterizing MKN-45 as CEA-high (CEA_high_) [30], making it a suitable positive control for this proof-of-concept study. It must be noted that alternative cell lines, including colorectal cancer (CRC)-specific cell lines, could also be tested in the proposed assay to evaluate the targeting efficacy of CEA TCBs in various tumor types.

The rectal organoid line showed the second-highest CEA expression, while the small intestinal organoid line demonstrated CEA levels lower than the rectal organoid but still higher than DLD-1, which had the lowest CEA expression overall (CEA_low_) (Figure 1D). The two selected organoid lines were subsequently used for the co-culture and 3D-RDL assays to further evaluate the efficacy and safety of the CEA-targeting bispecific antibody *Cibisatamab*.

### 3.2. Probing Assay Specificity and Spontaneous Cytotoxicity of Allogeneic hPBMCs

To replicate the tumor-surrounding immune cell compartment, we utilized allogeneic hPBMCs obtained from four different healthy donors. For this study, the whole PBMCs were selected to replicate the intricate cellular environment of the immune system, enabling natural interactions between T cells and antigen-presenting cells (APCs), such as monocytes and dendritic cells. This approach offered a more physiologically relevant model for studying immune responses, as it closely mimicked the in vivo dynamics of immune cell interactions. It is important to note that isolated T cells or specific CD4/CD8 subpopulations could be employed to achieve more precise effector-to-target ratios and minimize potential interference from other cell types. Furthermore, autologous PBMCs matched to the organoid tissue could enhance the relevance of the experimental system; however, these approaches were beyond the scope of this proof-of-concept study.

Prior to establishing the 3D-RDL assay, we cultured hPBMCs alone to measure their intrinsic ATP production and lactate dehydrogenase (LDH) release, validating the specificity of our readout. All tested hPBMCs exhibited low levels of ATP and LDH compared to the target cells, confirming that this study primarily detected target cell death during the 3D-RDL assay (Appendix A).

To evaluate any potential spontaneous killing resulting from an allogeneic reaction, we co-cultured healthy human rectal or small intestinal organoids, as well as DLD-1 and MKN-45 spheroids, with the allogeneic hPBMCs. After 72 h, we found that the hPBMCs did not induce significant cell death in the healthy human rectal and small intestinal organoids, DLD-1, or MKN-45 spheroids. None of the hPBMCs caused significant depletion of ATP, LDH release, or cell death (Appendix A). Only hPBMCs derived from one of the healthy donors (Donor_1) showed a slight increase in LDH release, leading to the person’s exclusion from further screenings in the presence of the compound.

### 3.3. Cibisatamab-Induced Selective Killing of CEA_high_ Expressing Cancer Spheroids

To evaluate the efficacy and safety of the CEA TCB *Cibisatamab*, we established the 3D-RDL assay (Figure 1A). In this assay, we first co-incubated *Cibisatamab* with 3D cancer spheroids derived from the adenocarcinoma cell lines MKN-45 and DLD-1. Our findings revealed that *Cibisatamab* showed high and selective potency against CEA-high-expressing MKN-45 (Figure 2A–D).

As a first indication for a selective cytotoxic effect of *Cibisatamab* on CEA_high_ cells, we observed a dose-dependent increase in LDH release from MKN-45 spheroids after 72 h treatment with escalating concentrations of *Cibisatamab*. LDH is a stable enzyme that is present in all cell types. It is rapidly released into the cell culture medium when the plasma membrane is damaged [31]. The rise in LDH levels in the supernatant is a direct indicator of increased target cell death, reflecting *Cibisatamab’s* effectiveness in inducing cell lysis in CEA-high-expressing cells. In contrast, LDH release from DLD-1 spheroids did not show a significant increase, except at the highest concentration of CEA TCB (Figure 2A).

Additionally, ATP content was measured after 72 h of the 3D-RDL assay. Apoptotic cells stopped synthesizing ATP and ATP levels quickly degraded. Therefore, ATP was a reliable marker of cell viability [32]. We found a decrease in ATP levels in MKN-45 spheroids with higher doses of *Cibisatamab*, indicating reduced cellular viability and increased cell death. In comparison, there was no significant reduction in ATP content for DLD-1 spheroids, underscoring the selective impact of *Cibisatamab* on CEA_high_ cells (Figure 2B).

Cell death was further corroborated by live/dead staining analysis, where the ethidium intensity markedly increased in MKN-45 spheroids even at the lowest concentration of *Cibisatamab*. This increase in ethidium staining indicated a significant increase in cell death, which continued to escalate with higher antibody concentrations, reinforcing the potent killing effect of *Cibisatamab* on CEA-expressing tumor cells (Figure 2C).

The direct comparison between MKN-45 and DLD-1 revealed that the cytotoxic effect of *Cibisatamab* on MKN-45 was significantly higher than on DLD-1, as demonstrated by ATP production, LDH release, and live/dead staining. Specifically, MKN-45 cells treated with 10 µM of *Cibisatamab* showed significantly greater cytotoxicity compared to all treatment doses applied to the DLD-1 cell line (Appendix A).

To complement these findings, we analyzed the size of each spheroid using our image-based analysis pipeline. MKN-45 spheroids showed a noticeable decrease in size in response to increasing concentrations of *Cibisatamab*, which correlated with the observed reduction in viability. Conversely, the size of DLD-1 spheroids remained largely unaffected by *Cibisatamab* treatment, highlighting the minimal impact on cells with low CEA expression (Figure 2D).

Taken together, these results from the 3D-RDL assay provide a comprehensive demonstration of *Cibisatamab’s* selective targeting and killing of CEA_high_ expressing cells, such as MKN-45, while sparing the CEA_high_ DLD-1 spheroids.

Next, we showed that the selective killing of CEA_high_-expressing MKN-45 spheroids correlated with increased T cell activation upon *Cibisatamab* treatment. We investigated the T cell activation markers CD69, for early activation, and CD25, for late activation, by flow cytometry after 72 h co-culture of CEA_high_ MKN-45 spheroids with *Cibisatamab* and hPBMCs [32,33,34].

Due to the different kinetics of CD69 and CD25, we first analyzed their expression levels separately (Figure 3A,B) before evaluating the overall activation status of T cells by examining their co-expression (Figure 3C). To address potential overlap, we quantified T cells co-expressing CD69 and CD25, confirming that their co-expression increased with higher doses of *Cibisatamab* on both CD4+ and CD8+ T cell subpopulations (Figure 3C). This co-expression highlighted that T cells could simultaneously exhibit early and late activation states under appropriate stimulation conditions.

Overall, *Cibisatamab* induced a dose-dependent activation of both CD4+ and CD8+ T cell subpopulations, as evidenced by increased expression of CD25 and CD69 at the T cell surface (Figure 3A–C). Notably, CD8+ T cells showed higher upregulation of activation markers compared to CD4+ T cells.

### 3.4. Cibisatamab-Induced Killing of Healthy Human Rectal Organoids

To evaluate the safety profile and potential on-target, off-tumor effects of the CEA TCB *Cibisatamab*, we expanded our 3D-RDL assay to include healthy organoid models, specifically rectal and small intestinal organoids. These models allowed us to assess the impact of *Cibisatamab* on non-cancerous tissues with varying levels of CEA expression, providing insight into the selectivity of the antibody.

Our results showed that high doses of *Cibisatamab* induced cell death in rectal organoids, which exhibited intermediate CEA expression levels (Figure 1C,D). The decrease in organoid viability was evident from the decrease in ATP levels of the organoids after 72 h of treatment in the 3D-RDL assay. ATP levels in the rectal organoids with intermediate CEA expression significantly decreased with the three highest doses of *Cibisatamab*, correlating with increased cell death. In contrast, no significant reduction in ATP content was observed in the small intestinal organoids, reinforcing the selectivity of *Cibisatamab* against cells with higher CEA expression (Figure 4A). Additionally, we observed a dose-dependent increase in ethidium intensity following live/dead staining in rectal organoids with increasing concentrations of *Cibisatamab* (Figure 4B). In comparison, the small intestinal organoids, which had lower CEA expression levels, did not show significant cell death even at the highest CEA TCB concentrations, indicating a higher resistance to off-tumor effects (Figure 4B). The direct comparison between rectal organoids and small intestinal organoids showed that the cytotoxic effect of *Cibisatamab* on rectal organoids was significantly higher than its effect on small intestinal organoids (Appendix A). In the ATP assay, the ATP production in rectal organoids treated with 10 µM of *Cibisatamab* was significantly lower compared to the three highest treatment doses in small intestinal organoids. Additionally, for the live/dead staining, the cytotoxic effect on rectal organoids treated with 10 µM of *Cibisatamab* was significantly higher compared to all doses tested for the small intestinal organoids.

The decrease in rectal organoid viability could also be detected in brightfield images of rectal organoids co-incubated with hPBMCs and *Cibisatamab* (Figure 4D); however, the reduced viability was not captured in the organoid size analysis using our image-based analysis pipeline. Neither the rectal nor the small intestinal organoids exhibited significant size changes upon *Cibisatamab* treatment, suggesting that the antibody’s effect on organoid size was less pronounced in non-cancerous tissues (Figure 4C).

To directly compare the potency of the CEA TCB across the different cancer spheroids and healthy organoid lines, we analyzed the specific lysis after 72 h in the 3D-RDL assay, quantified by the ATP content, between the rectal and intestinal healthy organoids and the 3D cancer spheroids derived from MKN-45 and DLD-1. *Cibisatamab* selectively induced specific lysis in the CEA_high_-expressing MKN-45 cancer spheroids, as well as milder specific lysis in rectal organoids with intermediate CEA expression, while sparing the small intestinal organoids and the DLD-1 spheroids, both of which had low CEA levels (Figure 4E).

These findings strongly support the specificity of CEA TCB *Cibisatamab* for targeting CEA-expressing cells, demonstrating its capacity to induce cell death in CEA_high_- and intermediate CEA-expressing tissues, with limited off-tumor cytotoxicity in healthy tissues that express low levels of CEA.

### 3.5. Granzyme B Induction and Cytokine Release upon Cibisatamab Treatment

Next, we analyzed the release of Granzyme B in the supernatant collected from the 3D-RDL assay, which included the co-cultured hPBMCs together with MKN-45 and DLD-1 cancer spheroids, as well as healthy rectal and intestinal organoid lines, treated at two different doses (10 and 100 nM) of *Cibisatamab* for 72 h. Our results demonstrated a dose-dependent increase in Granzyme B levels across the tested models (Figure 5). At the higher dose (100 nM), hPBMCs co-cultured with MKN-45 spheroids and rectal organoids showed a pronounced increase in Granzyme B release compared to co-cultures with CEA_low_ DLD-1 spheroids and small intestinal organoids (Figure 5A). Granzyme B, a serine protease, played a pivotal role in inducing target cell apoptosis by facilitating cytotoxic T cell- and natural killer (NK) cell-mediated cytolytic activity [35]. The elevated Granzyme B release observed in the MKN-45 spheroids and rectal organoids aligned with the increased cell death induced by *Cibisatamab* in these models, highlighting a robust CEA TCB-mediated immune response. To further investigate this response, we examined Granzyme B release in the CEA_high_ MKN-45 spheroid co-culture across a range of *Cibisatamab* doses (0.001–100 nM) (Figure 5B). The dose-dependent increase in Granzyme B secretion corresponded with the cytotoxic effects of *Cibisatamab* observed at the three highest concentrations (compare Figure 2A–D).

Additionally, we quantified the soluble cytokines TNF-α and IFN-γ in the supernatants of MKN-45 co-cultures (Appendix A). A dose-dependent increase was observed for both cytokines, with *Cibisatamab* inducing high levels at the two highest concentrations. This increase corresponded to the granzyme B secretion and apoptosis observed in MKN-45 spheroids. TNF-α and IFN-γ were key mediators of inflammatory responses, playing a central role in driving intestinal inflammation and epithelial damage, underscoring the immune activation induced by *Cibisatamab* [36].

Taken together, these results highlight the importance of Granzyme B, IFN-γ, and TNF-α as key players in the immune response triggered by *Cibisatamab*. Together, they provide insights into *Cibisatamab’s* ability to activate the immune system and kill CEA_high_-expressing tumor cells effectively.

## 4. Discussion

In this study, we developed and validated a 3D redirect lysis (3D-RDL) assay to comprehensively evaluate the potency and safety of CEA TCB antibodies. We used this assay to evaluate *Cibisatamab*, demonstrating its selective cytotoxic effects on CEA_high_ expressing cancer cells while minimizing on-target, off-tumor activity on healthy tissues. The integration of this assay with various complementary analytical techniques, including bulk RNA sequencing, spectral flow cytometry, supernatant analysis, and high-throughput imaging, highlighted the versatility of our preclinical pipeline in dissecting the complex interactions between immune cells and tumor cells.

Our results revealed that *Cibisatamab* exhibited significantly higher potency against MKN-45 cancer spheroids, characterized by high CEA expression, compared to DLD-1 spheroids with lower CEA levels. This selective killing was confirmed through multiple readouts, including LDH release, ATP depletion, and image-based viability analysis, all of which consistently showed increased cell death in MKN-45 spheroids with escalating concentrations of *Cibisatamab*. Notably, MKN-45 spheroids also displayed a decrease in size, further supporting the correlation between antibody treatment and tumor cell viability reduction. In contrast, DLD-1 spheroids exhibited minimal changes across these metrics, underscoring the specificity of *Cibisatamab* for cells with CEA_high_ expression.

CEA TCB treatment led to the dose-dependent activation of both CD4+ and CD8+ T cells, measured through the upregulation of CD69 and CD25, with CD8+ T cells showing a more pronounced activation response. This finding aligns with the expected mechanism of action for TCBs, where T cell engagement is essential for the induction of targeted cytotoxicity.

The extension of our 3D-RDL assay to healthy organoid models allowed us to evaluate the on-target, off-tumor effects of *Cibisatamab*. We found that rectal organoids, which have intermediate CEA expression, showed a dose-dependent increase in cell death upon treatment, while small intestinal organoids, with low CEA levels, were not sensitive to the effects of CEA TCB. These observations were substantiated by ATP depletion and increased ethidium intensity in the rectal organoids but not in the small intestinal organoids. Importantly, the absence of significant size reduction in either organoid type suggests that *Cibisatamab’s* impact on non-cancerous tissues might be limited to cell death without a higher impact on tissue viability.

When directly comparing the cytotoxic effects of *Cibisatamab* across cancer spheroids and healthy organoids, we observed a clear trend of specific lysis in CEA_high_ MKN-45 cancer spheroids and rectal organoids with intermediate CEA expression, while sparing DLD-1 spheroids and small intestinal organoids with lower CEA levels. This selective action supports the hypothesis that *Cibisatamab* predominantly targets cells with high CEA expression, thus reducing the risk of off-tumor toxicity in healthy tissues. Such specificity is crucial for the clinical application of TCBs, as it underlines the importance of targeting markers that are highly expressed on tumors but minimally present on normal tissues to minimize adverse effects.

Our analysis further revealed a dose-dependent increase in Granzyme B and cytokine release, particularly in CEA_high_ MKN-45 spheroids and rectal organoids. The elevated Granzyme B secretion strongly correlated with enhanced T cell-mediated cytotoxicity observed in these models, highlighting the robust immune activation induced by *Cibisatamab*. The pronounced Granzyme B induction in CEA_high_ models underscored its critical role as a key mechanism driving the therapeutic efficacy of *Cibisatamab*, distinguishing these models from CEA_low_ spheroids and organoids where the response was markedly less potent. Additionally, the dose-dependent increase in TNF-α and IFN-γ levels correlated with Granzyme B release and the observed apoptotic effects in MKN-45 spheroids, underscoring the strong immune activation induced by *Cibisatamab*.

These results validate our 3D-RDL assay as a tool for preclinical testing, enabling the evaluation of therapeutic candidates. This model offers advantages over traditional 2D co-culture systems, which often fail to capture the complex dynamics of immune–tumor interactions in a physiologically relevant manner. By replicating the 3D architecture of tissues and allowing for analysis of cellular responses, our platform provides a prediction of the efficacy and safety profiles of immunotherapies, thus aiding in the identification and optimization of therapeutic leads.

## 5. Conclusions

In conclusion, our study establishes the 3D-RDL assay as a preclinical tool for assessing the safety and potency of immune-modulatory agents, with a focus on CEA-targeting therapies like *Cibisatamab*. The integration of diverse analytical techniques within our pipeline—encompassing RNA sequencing, flow cytometry, viability assays, and high-throughput imaging—enables a multi-dimensional evaluation of both tumor and healthy tissue responses. This approach not only allows for precise characterization of on-target cytotoxic effects but also provides critical insights into potential off-tumor liabilities, enhancing the predictive accuracy of preclinical safety profiles.

Our findings with *Cibisatamab* illustrate its selective targeting capability, with potent killing of CEA-high-expressing cancer cells and minimal effects on healthy tissues with low CEA expression. This specificity is crucial for reducing on-target, off-tumor toxicity and improving the therapeutic window of CEA TCBs in clinical settings.

Beyond CEA TCBs, our adaptable platform is well suited for evaluating a wide array of immunotherapies, including immune checkpoint inhibitors, NK cell engagers, innate immune activators, and combination therapies, as well as cellular immunotherapies such as CAR-Ts and tumor-infiltrating lymphocytes (TILs).

This proof-of-concept study offers several avenues for extending the 3D-RDL assay to further enhance its translational relevance. For example, the use of tumoroids derived from colorectal cancer (CRC) patients together with autologous PBMCs or T cells will provide a more patient-specific model for studying immune–tumor interactions and treatment efficacy. Additional assay readouts can further enhance the study relevance. Live cell imaging and TIL mobility assays can be applied to directly analyze interactions between spheroids/organoids and T cells or TILs upon treatment. Moreover, readouts, such as Caspase-3 staining, could be incorporated to specifically monitor epithelial-cell or cancer-cell apoptosis.

The versatility of our assay provides an asset for preclinical research, offering robust datasets to support the development and optimization of next-generation cancer immunotherapies. Our approach not only facilitates the identification of promising therapeutic candidates but also aids in refining dosing strategies, ultimately contributing to the design of safer and more effective treatment options for patients with solid tumors.

## 6. Patents

The Ecole Polytechnique Fédérale de Lausanne has filed for patent protection on the technology described herein (WO2018/050862), and S.H. and N.B. are named as inventors on this patent; S.H. and N.B. are shareholders in Doppl SA, who is commercializing this patent.

## Figures and Tables

**Figure 1 cancers-17-00291-f001:**
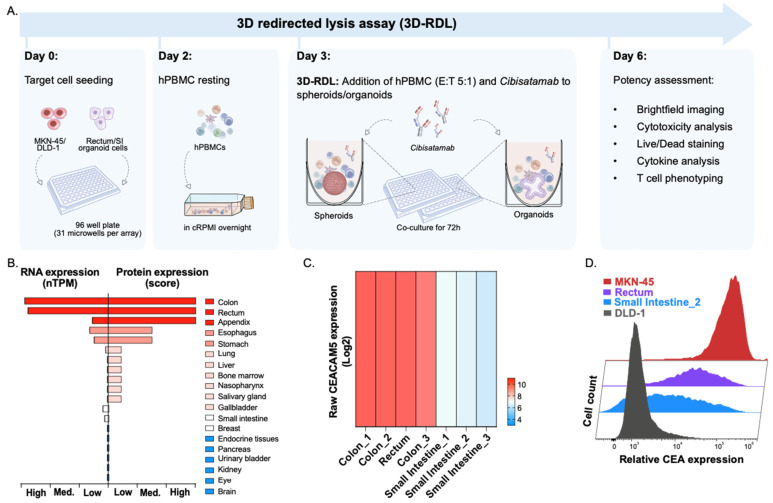
The experimental design of the 3D-RDL assay and CEA target expression validation: (**A**) In the preparation of the 3D-RDL assay, single cells from cell lines (MKN-45 and DLD-1) and organoids (Rectum and Small intestine_2) are seeded in 96-well microwell plates. At day 2, hPBMCs are rested overnight in cRPMI medium. At day 3, hPBMCs (E:T, 5:1) and *Cibisatamab* (six doses; 0.001–100 µM) are added to the aggregated spheroids and organoids and incubated for 72 h. (**B**) CEA protein and RNA expression levels in healthy human organs. CEA is expressed in a variety of epithelial tissues such as the urogenital, respiratory, and gastrointestinal tracts. In the healthy human colon, the CEA protein is restricted to the apical side of the differentiated epithelial cells forming the luminal surface. Source: Human Protein Atlas. (**C**) CEACAM5 bulk RNA expression levels in healthy colon, rectal, and intestinal organoids (Doppl SA Biobank). (**D**) Relative CEA protein expression levels of control cell lines MKN-45 (CEA_high_) and DLD-1 (CEA_low_) and 2 different healthy organoid lines (Rectum and Small intestine_2) analyzed by flow cytometry.

**Figure 2 cancers-17-00291-f002:**
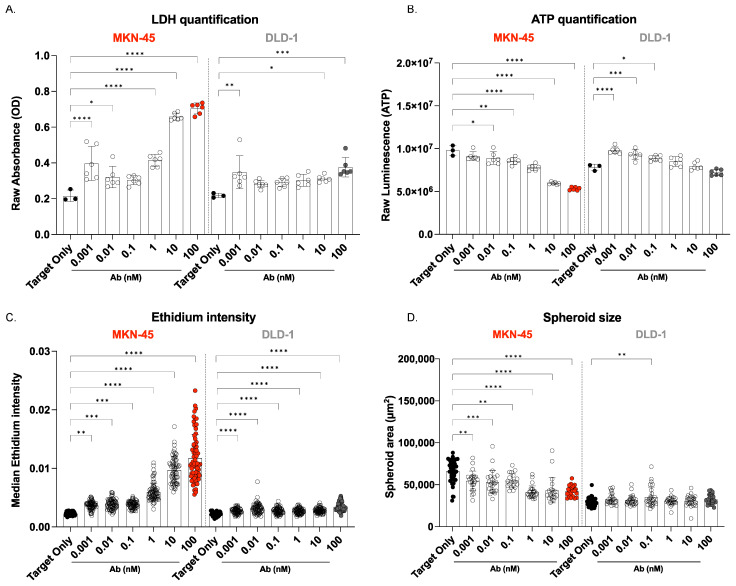
The 3D-RDL results of co-cultured 3D grown human adenocarcinoma cell lines MKN-45 and DLD-1 in the presence of hPBMCs treated at different doses of *Cibisatamab* for 72 h. (**A**–**D**) Cytotoxicity of MKN-45 and DLD-1 cells in the presence of *Cibisatamab* and healthy hPBMCs (E:T,5:1) for 72 h in a 3D-RDL assay. Each well was composed of 31 microwells containing one organoid/spheroid each. (**A**) Lactate dehydrogenase (LDH) released in the supernatant was quantified, and raw absorbance values was displayed. Increasing concentrations of LDH reflect an increase in target cell death. (**B**) ATP content was measured through luminescence release upon complete cell lysis. Decreasing ATP concentrations show increase in target cell death. (**C**) Ethidium median intensity was evaluated with our high-throughput automated analysis pipeline (imaged based). Increased ethidium signals correlated with increased cell death. (**D**) Our image-based analysis pipeline enabled the measurement of the size of each organoid. A decrease in spheroid size generally correlated with a decrease in cell viability. Data represent mean ± SD from 3 different hPBMCs donors tested in duplicate. Ordinary 1-way ANOVA with Dunnett’s multiple-comparisons test was performed, and the mean of each column was compared to the mean of the reference column (Target only). * *p* < 0.05, ** *p* < 0.01, *** *p* < 0.001, and **** *p* < 0.0001. Only significant values are displayed (*p* < 0.05).

**Figure 3 cancers-17-00291-f003:**
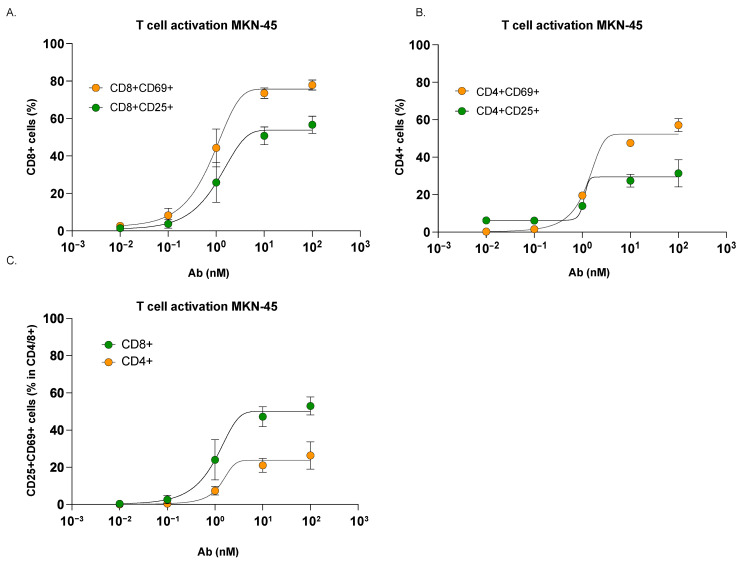
T cell activation after 72 h of co-culture of hPBMCs and MKN-45 cells in 3D-RDL assay format. (**A**) CD8+ and (**B**) CD4+ T cell response in the presence of healthy hPBMCs (E:T, 5:1), MKN-45 cells and increasing doses of CEA TCB antibodies. (**C**) CD8+ and CD4+ co-expression in the presence of healthy hPBMCs (E:T, 5:1), MKN-45 cells, and increasing doses of *Cibisatamab*. T cell activation upon *Cibisatamab* treatment was measured by flow cytometry while measuring the expression of CD25 (green) and CD69 (orange). Data represent non-linear regression analysis of the mean ± SD from 3 different hPBMCs donors tested in duplicate.

**Figure 4 cancers-17-00291-f004:**
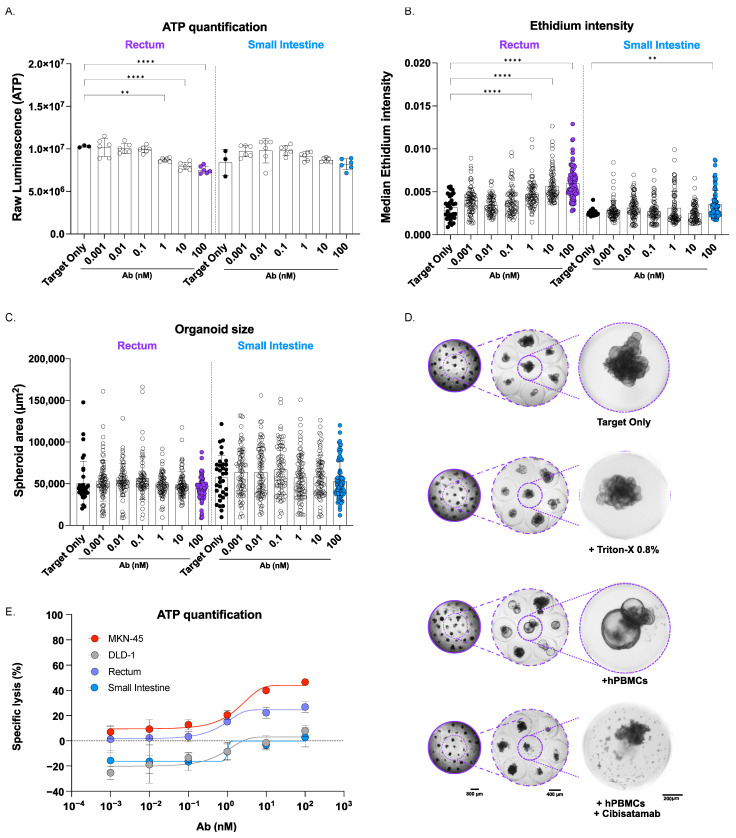
The 3D-RDL results of co-cultured rectal and intestinal healthy organoids in the presence of hPBMCs treated at different doses of *Cibisatamab* for 72 h. (**A**–**C**) Cytotoxicity of healthy rectal and intestinal organoids in the presence of *Cibisatamab* and healthy hPBMCs (E:T, 5:1) for 72 h in a RDL assay. Each well was composed of 31 microwells containing one organoid/spheroid each. (**A**) Ethidium median intensity was evaluated with our high-throughput automated analysis pipeline (imaged based). Increased ethidium signals correlated with increased cell death. (**B**) Our image-based analysis pipeline enabled the measurement of the size of each organoid. Decreases in spheroid size generally correlated with decreases in cell viability. (**C**) ATP content was measured through luminescence release upon complete cell lysis. Decreasing ATP concentrations showed increases in target cell death. (**D**) Representative brightfield images of rectal organoids cultured in microwells in different conditions after 48 h: untreated rectal organoids (target cells alone); rectal organoids treated with Triton-X 0.8% (+Triton-X 0.8%); rectal organoids co-incubated with hPBMCs; rectal organoids co-incubated with hPBMCs and 100 nM of *Cibisatamab*. Scale bars: 800 μm, 400 μm, and 200 μm. (**E**) The cytotoxicity of cancer spheroids and healthy organoids in the presence of *Cibisatamab* and healthy hPBMCs (E:T, 5:1) for 72 h in a 3D-RDL assay. Data represent non-linear regression analysis of the mean ± SD from 3 different hPBMCs donors tested in duplicates. Specific lysis was calculated as follows: ((Sample—Target Only)/(Triton-X 0.8%—Target Only)). Data represent mean ± SD from 3 different hPBMCs donors tested in duplicate. Ordinary 1-way ANOVA with Dunnett’s multiple-comparisons test was performed, and the mean of each column was compared to the mean of the reference column (Target only). ** *p* < 0.01, and **** *p* < 0.0001. Only significant values are displayed (*p* < 0.05).

**Figure 5 cancers-17-00291-f005:**
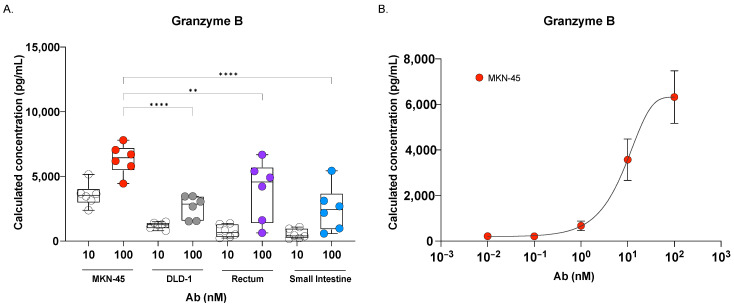
Granzyme B quantification after 72 h of co-culture of rectal/intestinal healthy organoids and 3D grown MKN-45/DLD-1 spheroids in the presence of hPBMCs treated at different doses of CEA TCB. (**A**,**B**) The quantification of Granzyme B released in the supernatant after 72 h RDL assay by flow cytometry. Data represent non-linear regression analysis of the mean ± SD from 3 different hPBMCs donors tested in duplicate. Ordinary 1-way ANOVA with Dunnett’s multiple-comparisons test was performed, and the mean of each column compared to the mean of MKN-45 treated at 100 nM. ** *p* < 0.01 and **** *p* < 0.0001. Only significant values are displayed (*p* < 0.05).

## Data Availability

The data that support the findings of this study are available on reasonable request from the authors.

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
