# Peer review of "In Vitro Evaluation of the Safety and Efficacy of Cibisatamab Using Adult Stem Cell-Derived Organoids and Colorectal Cancer Spheroids"

_cancers, 2025, doi:10.3390/cancers17020291_

Round 1

Reviewer 1 Report

Comments and Suggestions for Authors

The authors proposed a tumor cell-immune cell co-culture 3D model system to evaluate the safety and efficacy of bispecific antibodies. The authors showed the on-target and off-target effects of Cibisatamab, a bispecific antibody targeting CEA and CD3, tested using the 3D model system.

Comments and suggestions:

1.        Sup Fig1. C-E are missing.

2.        Fig 2 A-D and line 273: “Cibisatamab had higher potency against MKN-45 cells compared to DLD-1 cells”

·      Stats showed significances in MKN-45 and DLD-1 treated by different concentration of antibodies. A comparison between MKN-45 and DLD-1 would be necessary to state that Cibisatamab had higher potency against MKN-45 cells compared to DLD-1 cells.

3.        Need reference for using CD25 as a marker for “late” T cell activation. The expression of CD69 and CD25 are not mutually exclusive on T cells. What are the reasons gating T cells using CD69 and CD25 separately. Authors would need to show if there is an overlap between CD69+ T cell and CD25+ T cells, and the objectives of gating T cells by CD69 and CD25 separately.

4.        Line 342: Fig 1D is missing.

5.        Fig. 5A needs a cleanup on unnecessary comparisons.

Author Response

We appreciate the feedback provided by Reviewer 1. We have now addressed their comments and suggestions.

  1. Sup Fig1. C-E are missing.

We apologize for this mistake. Suppl. Fig. 1 C-E does not exist and the figure we refer to in line 348 is now called “Suppl. Fig. 3A-C”.

  1. Fig 2 A-D and line 273: “Cibisatamab had higher potency against MKN-45 cells compared to DLD-1 cells”. Stats showed significances in MKN-45 and DLD-1 treated by different concentration of antibodies. A comparison between MKN-45 and DLD-1 would be necessary to state that Cibisatamab had higher potency against MKN-45 cells compared to DLD-1 cells.

We appreciate the reviewer’s comment highlighting the need for a direct comparison and statistical analysis between the treated MKN-45 and DLD-1 cell lines. In Section 3.3 (lines 377–381), we have added a detailed comparison of ATP production, LDH release, and ethidium staining between the two cell lines. Furthermore, we have included Supplementary Table 1 in the supplementary data, presenting the statistical analysis comparing MKN-45 cells treated with 10 µM Cibisatamab to all treatment conditions for DLD-1.

To ensure thoroughness, we have also incorporated a direct comparison between the rectal organoid line and the small intestinal organoid line. This has been added to Section 3.4 (lines 453–460) and supported by the inclusion of Supplementary Table 2 in the supplementary data.

  1. Need reference for using CD25 as a marker for “late” T cell activation. The expression of CD69 and CD25 are not mutually exclusive on T cells. What are the reasons gating T cells using CD69 and CD25 separately. Authors would need to show if there is an overlap between CD69+ T cell and CD25+ T cells, and the objectives of gating T cells by CD69 and CD25 separately.

We appreciate the reviewer’s insightful comment regarding the use of CD25 as a marker for late T cell activation and the relationship between CD69 and CD25 expression. To address this, we have added references supporting the use of CD25 as a late activation marker. Specifically, we included:

  • Antas PR, et al., Kinetics of T cell-activation molecules in response to Mycobacterium tuberculosis antigens, Mem Inst Oswaldo Cruz (2002), which describes the kinetics of CD25 expression on activated T cells.

  • Naghizadeh M, et al., Kinetics of activation marker expression after in vitro polyclonal stimulation of chicken peripheral T cells, Cytometry (2022), which analyzes CD25 expression after up to 120 hours of in vitro incubation.

These references have been incorporated as references 33 and 34 in Section 3.3, line 412.

Additionally, we performed an overlap analysis between CD69+ T cells and CD25+ T cells, as suggested. The results of this analysis have been added to the revised manuscript (Section 3.3, Figure 3C, and lines 413–419), providing clarity on the gating strategy and demonstrating the objectives of analyzing CD69 and CD25 expression separately.

  1. Line 342: Fig 1D is missing.

We apologize for the oversight. The CEA levels referenced in the revised manuscript are presented in Figure 1C (RNA) and Figure 1D (Protein). Compare section 3.1, line 288 and line 308.

  1. 5A needs a cleanup on unnecessary comparisons.

We agree with the reviewer’s observation and have revised Figure 5A accordingly. The updated figure now focuses on the relevant comparison, specifically between the 100 nM antibody treatment of MKN-45 (highest Granzyme B release) and DLD-1, rectal organoids, and small intestinal organoids at the same concentration. This adjustment ensures a clearer and more targeted presentation of the data.

Reviewer 2 Report

Comments and Suggestions for Authors

After reviewing the article entitled "Evaluating the Safety and Efficacy of T Cell Bispecific Antibodies with Adult Stem Cell-derived Organoids and Colorectal Cancer Spheroids" authored by Anstett et al, and submitted for consideration as a potential publication in Cancers, as a reviewer of this important work, I would like to offer the following:

General opinion:

The study presents a highly relevant and innovative approach to the evaluation of immunotherapies, particularly T cell bispecific antibodies, using complex 3D models that better mimic in vivo conditions. It makes a significant contribution to the field of cancer immunotherapy by improving preclinical testing, predicting therapeutic responses, and identifying potential risks such as off-target effects. The methodology and results are valuable for advancing the development of next-generation immunotherapies with improved specificity and safety profiles.

Suggested modifications:

- Please consider changing the title of your study to the following: In vitro evaluation of cibisatamab using stem cell-derived organoids and colorectal cancer spheroids.

- Please include a figure showing the design of the study.

- Describe the statistical methods used in your study.

Author Response

We would like to thank Reviewer 2 for for the overall positive feedback. We have addressed their minor comments and suggestions.

  1. Please consider changing the title of your study to the following: In vitro evaluation of cibisatamab using stem cell-derived organoids and colorectal cancer spheroids.

We like this suggestion. We have revised the title of our study to: In vitro Evaluation of the Safety and Efficacy of Cibisatamab using Adult Stem Cell-derived Organoids and Colorectal Cancer Spheroids.

  1. Please include a figure showing the design of the study.

We appreciate the suggestion and have included a schematic in Figure 1A to illustrate the design of the study. We would also like to note, that we also adjusted Figure 1C and reduced the heatmap to only the seven relevant organoid lines.

  1. Describe the statistical methods used in your study.

We apologize for the oversight. We have now included a section describing the statistical analysis methods used in the study (compare lines 260-274).

Reviewer 3 Report

Comments and Suggestions for Authors

In this study, the authors developed a 3D redirect lysis model (3D-RDL) based on colonic cancer spheroids and patient-derived intestinal organoids to evaluate the potency/safety of Cibisatamab, a bispecific antibody targeting CEA on cancer cells and CD3 on T cells. This model, coupled with various complementary techniques, can be a relevant and reliable tool to predict the efficacy and safety of a wide range of immunotherapies.

However interesting and well designed, this study raised some concerns and questions.

Specific comments :

-Cibisatamab targets CEA on tumor cells and CD3 on T cells. Why do the authors used whole PBMCs in their expriments? It would have been more appropriate to isolate CD3 T lymphocytes and then CD4 / CD8 by magnetic sorting before co-culture with spheroids/organoids to be sure to have the exact E:T ratio for the subpopulation of T cells considered and also to get rid of other cell types (monocytes, granulocytes, B cells etc) that can interfere with the results.

-Granzyme B release in the supernatants of the 3D-RDL assay was measured using a Legendplex assay (CD8/NK panel). What about the other cytotoxic cytokines / products released (TNFa, IFNg, granzyme A, perforin), measurable in this assay? They should have been quantified. Are they increased in the same way as granzyme B in co-cultures with CEA-high spheroids / organoids + bispecific Ab?

-Organoids are derived from biopsies of colon and/or ? rectum, and small intestine (which region of the small intestine?). Please specify in the MM section the number of patients and their clinicopathological data. Are biopsies obtained during patients’ follow-up or at distance from cancer or other disease?

- MKN45 is a gastric adenocarcinoma cell line (gastric cancer is different from colorectal cancer (CRC)). The authors could have used several CRC cell lines with various CEA expression levels (e.g. LS174T and T84 express CEACAM5).

 -MM section, line 185 : “hPBMC were retrieved from the 3D-RDL assay, filtered and transferred…” : only cells in the supernatant were collected? Do immune cells also entered the spheroids/organoids?  Was videomicrosopy done to follow the kinetics with fluorescent markers?

-Cell death/viablility were monitored with LDH and ATP assays. Caspase-3 activity could have also been monitored to follow epithelial cells/cancer cells apoptosis.

-Allogeneic PBMC from healthy donors were used. It would be very interesting to perform autologous co-cultures if possible in the future (and to generate tumoroids from CRC patients co-cultured with autologous TILs or PBMCs).

-The organoids from normal colon/rectum and small intestine should have been characterized before the co-culture experiments. Please specify the various cell types present (enterocytes, goblet cells, Paneth cells, stem cells…) and if cell death occur specifically in a given cell type or not. In the MM section, lines 145-148: “full medium” : is it growth medium or differentiation medium? Please add some more information about the organoid cultures and media used.

Minor comments

Line 266: Suppl Fig 1C-E are missing in the suppl Fig 1.

Fig 2C.  DLD1: Please check the p values on the graph : very few differences in ethidium intensities between control and ab-treated spheroids and  p < 10-4 ?

Fig 4D: a higher magnification focusing on one representative well of each condition should also be shown ; the organoid structure is difficult to see and PBMC are not visible in the lower panel.

Author Response

We would like to thank Reviewer 3 for the detailed feedback, comments, and suggestions. We have carefully addressed all of their points and made the necessary revisions to improve the manuscript.

  1. Cibisatamab targets CEA on tumor cells and CD3 on T cells. Why do the authors used whole PBMCs in their expriments? It would have been more appropriate to isolate CD3 T lymphocytes and then CD4 / CD8 by magnetic sorting before co-culture with spheroids/organoids to be sure to have the exact E:T ratio for the subpopulation of T cells considered and also to get rid of other cell types (monocytes, granulocytes, B cells etc) that can interfere with the results.

We fully agree with the reviewer's comment and appreciate the suggestion. However, isolating specific T cell subsets was not within the scope of this proof-of-concept study. For this study, we used whole PBMCs to replicate the complex cellular environment of the immune system, enabling natural interactions between T cells and antigen-presenting cells (APCs), such as monocytes and dendritic cells. This approach provides a more physiologically relevant model for studying immune responses, as it closely mimics the in vivo dynamics of immune cell interactions. It is important to note that isolated T cells or specific CD4/CD8 subpopulations could be employed in future studies to achieve more precise effector-to-target ratios and reduce potential interference from other cell types. Additionally, the use of autologous PBMCs matched to the organoid tissue could further enhance the relevance of the experimental system. We have now adjusted section 3.2 (lines 328-337) and section 5 (line 613) to clarify our rationale and discuss these alternative approaches for similar studies.

  1. Granzyme B release in the supernatants of the 3D-RDL assay was measured using a Legendplex assay (CD8/NK panel). What about the other cytotoxic cytokines / products released (TNFa, IFNg, granzyme A, perforin), measurable in this assay? They should have been quantified. Are they increased in the same way as granzyme B in co-cultures with CEA-high spheroids / organoids + bispecific Ab?

We appreciate the reviewer's comment. Indeed, we also analyzed TNF-α and IFN-γ in the 3D-RDL and have now included these results in Supplementary Figures 4 and 5. Both cytokines showed a clear increase with higher Cibisatamab concentrations (Suppl. Fig. 5), indicating strong immune activation. This enhanced cytokine production occurred alongside the release of the cytotoxic mediator Granzyme B, highlighting the immune-stimulatory effects of Cibisatamab on hPBMCs. We have updated the manuscript to reflect these findings and provide a more comprehensive analysis of the immune response in our experimental setup (compare section 3.5, line 520-523).

  1. Organoids are derived from biopsies of colon and/or ? rectum, and small intestine (which region of the small intestine?). Please specify in the MM section the number of patients and their clinicopathological data. Are biopsies obtained during patients’ follow-up or at distance from cancer or other disease?

We have revised the methods section to provide the requested details regarding the patient samples used in this study. Biopsies were collected from a total of 7 patients (age 61 ± 16, 71% female), with normal adjacent tissue (NAT) obtained during surgical tumor resection in patients diagnosed with colorectal, ileal, or pancreatic cancers. Human intestinal crypts were isolated from biopsies taken from the rectum, colon (transverse, ascending, and appendix), and small intestine (jejunum and duodenum) according to established protocols. The updated information can be found in lines 140-145 of the manuscript.

  1. MKN45 is a gastric adenocarcinoma cell line (gastric cancer is different from colorectal cancer (CRC)). The authors could have used several CRC cell lines with various CEA expression levels (e.g. LS174T and T84 express CEACAM5).

We thank the reviewer for this valuable comment. We appreciate your suggestion regarding the use of additional colorectal cancer (CRC) cell lines. For this proof-of-concept study, we selected MKN-45, a gastric adenocarcinoma cell line that exhibits high and well-characterized CEA expression. This choice was based on MKN-45's established use in studies involving CEA-targeted therapies. However, we would like to acknowledge that there are several other CRC cell lines, such as LS174T and T84, that express varying levels of CEACAM5 and could also be utilized in similar studies to broaden the applicability of the findings.We also mentioned this in the revised manuscript (compare section 3.1, line 300-305)

  1. MM section, line 185 : “hPBMC were retrieved from the 3D-RDL assay, filtered and transferred…” : only cells in the supernatant were collected? Do immune cells also entered the spheroids/organoids? Was videomicrosopy done to follow the kinetics with fluorescent markers?

We thank the reviewer for this comment. Indeed, for the respective analysis, only the PBMCs present in the supernatant were collected. We agree that for follow-up studies, visualizing the kinetics of immune cells or TILs, for example through a TIL mobility assay, would add further relevance and more insights. We have mentioned this potential in section 5, lines 612-614.

  1. Cell death/viablility were monitored with LDH and ATP assays. Caspase-3 activity could have also been monitored to follow epithelial cells/cancer cells apoptosis.

For this proof-of-concept study, we selected the LDH and ATP assays, along with ethidium staining, to monitor cell death and viability. However, we agree that monitoring Caspase-3 activity to follow epithelial cell/cancer cell apoptosis would be valuable. We acknowledge that the panel of readouts can be expanded in similar follow-up studies, as mentioned in section 5, line 615.

  1. Allogeneic PBMC from healthy donors were used. It would be very interesting to perform autologous co-cultures if possible in the future (and to generate tumoroids from CRC patients co-cultured with autologous TILs or PBMCs).

We agree, this would be very insightful and we would like to mention that our proof-of-concept study offers several avenues for extending the 3D-RDL assay to enhance its translational relevance. For example, incorporating tumoroids derived from colorectal cancer (CRC) patients, in co-culture with autologous PBMCs or T cells, would provide a more patient-specific model for studying immune-tumor interactions and treatment efficacy. We have mentioned this potential extension in Section 5, lines 608-612.

  1. The organoids from normal colon/rectum and small intestine should have been characterized before the co-culture experiments. Please specify the various cell types present (enterocytes, goblet cells, Paneth cells, stem cells…) and if cell death occur specifically in a given cell type or not. In the MM section, lines 145-148: “full medium” : is it growth medium or differentiation medium? Please add some more information about the organoid cultures and media used.

We have now included a characterization of the organoid lines used in this study (Suppl. Figure 1) through immunofluorescence and bulk RNA sequencing, which demonstrate the expression of key gastrointestinal (GI) differentiation markers and cell types. However, in this study, we did not analyze which specific cell types were most affected by the treatment. As CEA is most highly expressed in the differentiated cells at the top of the crypts, we would expect these cells to be most affected by the treatment.

Additionally, we have provided more information regarding the organoid culture medium and clarified in the Methods section (lines 147-153).

  1. Line 266: Suppl Fig 1C-E are missing in the suppl Fig 1.

We apologize for this mistake. Suppl. Fig. 1 C-E does not exist and the figure we refer to in line 348 is now called “Suppl. Fig. 3A-C”.

  1. Fig 2C. DLD1: Please check the p values on the graph : very few differences in ethidium intensities between control and ab-treated spheroids and  p < 10-4 ?

TWe agree that the scale of the graph may make the p-values appear unexpectedly high. However, we have double-checked the analysis, and the values are correct. Due to the large sample size (3 replicates with 31 organoids/spheroids each), we quickly get such statistically significant results. We hope this clarifies the situation.

  1. Fig 4D: a higher magnification focusing on one representative well of each condition should also be shown ; the organoid structure is difficult to see and PBMC are not visible in the lower panel.

We have adjusted Figure 4D as suggested to include a higher magnification, focusing on one representative well of each condition. This revision provides a clearer view of the organoid structure, and the presence of PBMCs is now more visible in the panel with the highest magnification.

Round 2

Reviewer 3 Report

Comments and Suggestions for Authors

The authors have adequately addressed the comments and modified the manuscript and figures accordingly.

Minor comments :

-lines 557-558 : please mention supplemental Fig. 5, comment on the results obtained for IFNg and TNFa, and smooth the TNFa curve.

-supplemental Fig. 1B : lysozyme staining : it is difficult to distinguish specific staining from background. The positive cells could be identified by arrows or arrowheads.

Author Response

We have now addressed the additional comments provided by Reviewer 3.

Comment 1:

Lines 557-558 : please mention supplemental Fig. 5, comment on the results obtained for IFNg and TNFa, and smooth the TNFa curve.

We thank the reviewer for the helpful comment. In the revised manuscript, we have referenced and described Supplemental Figure 5 in the Results section (lines 519–525). Additionally, we included a sentence discussing the TNF-α and IFN-γ analysis in the Discussion section (lines 587–589). Lastly, we corrected the curve fitting for the TNF-α graph in Supplemental Figure 5 to enhance clarity.

Comment 2:

Supplemental Fig. 1B : lysozyme staining : it is difficult to distinguish specific staining from background. The positive cells could be identified by arrows or arrowheads.

We fully agree with this comment and have added white arrowheads to the lysozyme staining in Supplemental Figure 1B to make it easier to identify the positive cells.